# In vitro self-replication and multicistronic expression of large synthetic genomes

K. Libicher[1], R. Hornberger[1], M. Heymann [2] & H. Mutschler [1✉]

The generation of a chemical system capable of replication and evolution is a key objective of synthetic biology. This could be achieved by in vitro reconstitution of a minimal self-sustaining central dogma consisting of DNA replication, transcription and translation. Here, we present an in vitro translation system, which enables self-encoded replication and expression of large DNA genomes under well-defined, cell-free conditions. In particular, we demonstrate self-replication of a multipartite genome of more than 116 kb encompassing the full set of *Escherichia coli* translation factors, all three ribosomal RNAs, an energy regeneration system, as well as RNA and DNA polymerases. Parallel to DNA replication, our system enables synthesis of at least 30 encoded translation factors, half of which are expressed in amounts equal to or greater than their respective input levels. Our optimized cell-free expression platform could provide a chassis for the generation of a partially self-replicating in vitro translation system.

[1] Biomimetic Systems, Max Planck Institute of Biochemistry, Am Klopferspitz 18, 82152 Martinsried, Germany. [2] Intelligent Biointegrative Systems Group, University of Stuttgart, Pfaffenwaldring 57, 70569 Stuttgart, Germany. ✉email: mutschler@biochem.mpg.de

Self-encoded reproduction is a major requirement for the creation of artificial life[1]. In systems inspired by existing biochemistry, such as minimal protein-based cells (MPCs), self-replication demands a complete cell-free reconstitution of the central dogma of molecular biology, including DNA replication, transcription and translation[2–6]. In vitro protein synthesis from DNA can be achieved in well-defined recombinant systems based on phage RNA polymerases, core parts of the *Escherichia coli* translation machinery and a minimal energy regeneration system (PURE— Protein synthesis Using Recombinant Elements)[7]. In contrast, transcription–translation-coupled DNA replication (TTcDR) of a genome encoding all macromolecular components of the PURE system by a self-encoded replisome remains difficult[8]. DNA replication employing DNA polymerases (DNAP) from phages such as Phi29 are promising candidates for self-encoded TTcDR of minimal genomes[9,10]. For example, partially self-encoded TTcDR inside liposomes was accomplished using small linear Phi29 genomes encoding a minimal two-gene replicon on three kilobases (kb)[11]. TTcDR of the Phi29 full-length genome (~19 kb) in a PURE-based system was also achieved, but only if sufficient amounts of replication factors were either supplied externally or co-expressed from an excess of non-replicative DNA templates[11]. TTcDR of small circular DNAs (2 kb) encoding only the Phi29-DNAP was recently realised by coupling the reaction to Cre-Lox recombination[12]. Despite these advances, a concurrent, self-encoded replication and expression of the up to 150 genes (113 kb) proposed for MPC self-replication[3] is currently out of reach. Here, we describe a modified PURE reaction that enables direct co-expression and Phi29-DNAP-dependent TTcDR of large multicistronic DNA elements that reach the predicted genome size required to encode a minimal cell. In particular, we demonstrate self-replication of a multipartite genome larger than 116 kb encompassing the full set of *Escherichia coli* translation factors, all three ribosomal RNAs, an energy regeneration system, as well as RNA and DNA polymerases. Parallel to DNA replication, our system enables synthesis of at least 30 encoded translation factors, half of which are expressed in amounts equal to or greater than their respective input levels.

## Results

**PURErep enables self-encoded DNA replication**. Initially, we tested self-encoded Phi29-DNAP-dependent TTcDR using the standard protocol of the commercially available PURExpress system. The Phi29-DNAP coding region flanked by a T7 promoter was first cloned into a pCR-Blunt TOPO vector (pREP, Fig. 1a). In principle, this construct should enable spontaneous RNA-primed rolling-circle replication[13] by the self-encoded DNAP without additional replication proteins or externally supplied DNA primers as reported previously[10]. However, using the standard PURExpress reaction supplied with dNTPs and 4 nM pREP, we were unable to detect de novo synthesis of DNA by either agarose gel electrophoresis or qPCR (Fig. 1b, c). This finding is in agreement with previous studies which reported that the high tRNA and rNTP concentrations in standard PURE systems impair DNA-polymerase (DNAP) activity and that optimised custom systems are required to achieve efficient TTcDR[10,11]. In order to improve DNA replication without access to tailor-made PURE systems, we set out to optimise the PURExpress standard reaction protocol. To this end, we increased the relative amount of translation factors, ribosomes and reducing agent while decreasing tRNA and rNTP levels (Fig. 1d; Supplementary Table 1). Using this optimised PURE formulation (PURErep), we achieved, depending on the pREP input concentration, ~5–12-fold replication of pREP monomer

units in overnight TTcDR reactions (Fig. 1b, e). Full-length de novo synthesis of pREP was confirmed by MluI digestion of the replication product (Fig. 1c). Taking superfolder green fluorescent protein (sfGFP)[14] expression as an overall measure for in vitro translation (IVT) activity, we found that the altered PURE formulation resulted in a batch-dependent reduction of protein synthesis yields of 20–40% compared with the TTcDR-incompetent PURExpress system (Supplementary Fig. 1A, B). Thus, the improved compatibility of the PURErep system with DNA replication is achieved at the expense of only a modest reduction in overall protein expression strength.

**TTcDR products can be transformed and propagated in *E. coli***. A qPCR-based analysis of DNA replication revealed a robust doubling time of 1–2 h for different initial template concentrations with DNA replication proceeding even after 24 h at 30 °C (Fig. 1e). TTcDR of pREP was also sustainable for more than five successive generations of serial dilution when 4% of an overnight PURErep/pREP reaction was directly transferred into a fresh PURErep mix (Fig. 1f). This result implies that TTcDR products can serve as templates for self-coded DNA replication over several generations. As expected from the rolling-circle-type replication, we observed a considerable amount of product with low electrophoretic mobility, likely representing large molecular weight concatemers and/or DNA-MgPP$_i$ clusters as reported previously for similar reactions (Supplementary Fig. 1C)[15]. Unexpectedly, we also observed formation of ~5 kb products in unprocessed samples, suggesting that TTcDR reactions may produce considerable amounts of monomeric pREP copies (Supplementary Fig. 1C). We were also able to transform de novo synthesised products into *E. coli* after removal of parental plasmids (Supplementary Fig. 2A). Purified in vivo amplified products were identical in size to monomeric pREP (Supplementary Fig. 2B).

**PURErep enables TTcDR of large multipartite genomes**. Encouraged by the efficient TTcDR in PURErep, we set out to co-replicate a collection of genes coding for crucial components of the PURE reaction such as the 31 essential *E. coli* translation factors (TFs). To this end, we probed co-TTcDR of pREP (4.6 kb) together with each one of the three large plasmids pLD1 (30 kb, 13 translation factors – TFs), pLD2 (20 kB, 8 TFs), or pLD3 (23 kb, 9 TFs), which were recently cloned to enable recombinant expression of 30 of the 31 TFs[16]. Indeed, the TTcDR products of all four plasmids (including pREP) showed identical MluI restriction patterns as clonal plasmids conventionally propagated in *E. coli* (Fig. 2a). Moreover, the pLD TTcDR products could be directly transformed into *E. coli*, from where they were maintained as monomeric plasmids (demonstrated for pLD3, Supplementary Fig. 2C, D). The optimised PURErep mix enabled even the complete replication of all three pLD plasmids together with PURErep in a one-pot reaction (Fig. 2b; Supplementary Fig. 3A, B).

Next we sought to further expand the genetic load of the TTcDR-system by co-replicating plasmids encoding additional components of the PURE system such as EF-Tu (pEFTu), which is missing in the pLD system, and also the ribosomal RNA operon *rrnB* (prRNA), which encodes for 23S rRNA, 16S rRNA, 6S rRNA and tRNA(Glu2)[17] (Fig. 2c). qPCR experiments targeting plasmid-specific amplicons confirmed that monomer units of all six plasmids (total DNA length 93 kb) were replicated about 2–8-fold relative to their respective input levels in the presence of pREP and dNTPs after overnight incubation (Fig. 2c). In support of complete co-replication of all plasmids, transformations of DpnI-treated PURErep reaction products into *E. coli* resulted in colonies resistant to either zeocin (pREP), kanamycin (pLD plasmids and prRNA) or carbenicillin (pEFTu) (Fig. 2d). DNA preparations of 26 randomly

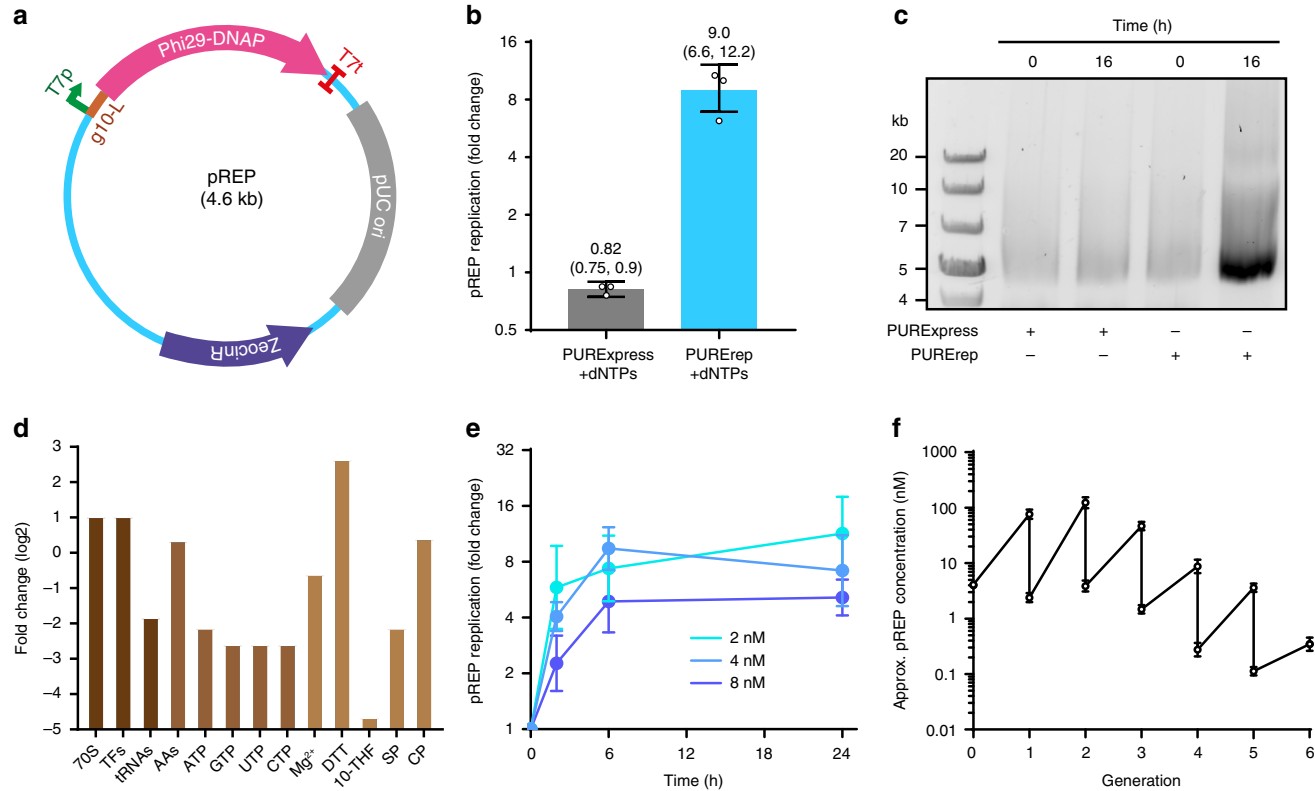

**Fig. 1 PURErep enables efficient transcription–translation coupled DNA replication. a** Map of pREP. The plasmid encodes the gene for Phi29-DNAP under the control of a T7 promotor (T7p) and a bidirectional T7-terminator (T7t). IVT of the DNAP gene is increased by a T7 g10-L leader sequence. A zeocin-resistance gene and a pUC origin allow selective propagation in *E. coli*. **b** Replication of pREP in PURExpress (grey) or PURErep (blue) after 6 h at 30 °C. Fold changes were determined by qPCR relative to the pREP input levels (4 nM). Bars show the means with their 68% confidence intervals (CI) from biological triplicates using different PURErep/PURExpress batches. **c** Image of representative agarose gels loaded with TTcDR samples of pREP (8 nM input DNA) after MluI treatment. Samples were tested in three biological replicates. **d** Relative changes in compound levels between PURErep and PURExpress (log$_2$-scale). Estimated compound levels for PURExpress are based on the numbers from Kuruma and Ueda[40] (TF translation factors, AAs amino acids, DTT dithiothreitol, 10-THF 10-Formyltetrahydrofolate, SP spermidine, CP creatine phosphate). **e** TTcDR of pREP at different input concentrations. Fold changes relative to input levels were measured by qPCR (means with 68% CI, biological triplicates using different PURErep batches for each concentration). **f** pREP propagation over repeated passages of serial transfer. After each overnight TTcDR reaction in PURErep, 4% of the volume was transferred into a freshly prepared, plasmid-free PURErep reaction. Fold changes relative to the initial concentration (4 nM) were used to approximate the concentrations before and after each generation (mean ± 68% CI, technical triplicates). Source data are available in the Source Data file.

picked clonal colonies followed by restriction pattern analysis indeed confirmed successful TTcDR of all six plasmids (Fig. 2e; Supplementary Fig. 3C–E). In contrast, almost no background colonies were obtained when samples from dNTP-free PURErep experiments were transformed into *E. coli* (Fig. 2d). Using the same approach (Fig. 3; Supplementary Fig. 4), we were able to demonstrate co-replication of five additional plasmids encoding all but one of the missing proteins of the PURE enzyme mix (Supplementary Table 2, except peptidylprolyl isomerase). The additional plasmids include the genes for a minimal nucleoside triphosphate regeneration system based on creatine kinase (pCKM), adenylate kinase (pAK1) and nucleoside diphosphate kinase (pNDK), as well as T7-RNA polymerase (T7RNAP) and pyrophosphatase (pIPP), which is added to more recent versions of the PURE system[18]. With a total size of 116.3 kb, this set of 11 plasmids reaches >100% of the predicted genome length proposed for a near-minimal, self-replicating system dependent only on small-molecule nutrients (Fig. 3a)[3].

**PURErep enables synthesis of 30 TFs during TTcDR.** Having shown combined TTcDR of the multicistronic plasmids that encode almost all proteins of the PURE enzyme mix, we explored

whether the PURErep mix could also enable parallel expression of these genes during replication. A (partially) self-replicating system based on the central dogma needs to be able to regenerate at least some of its different protein components. As a first step in this direction, we focused on the multicistronic expression of the TFs encoded on the three pLD plasmids pLD1, pLD2 and plD3 (not including pEFTu). To explore whether PURErep is generally capable of supporting multicistronic expression from these plasmids, we performed cell-free expression from each individual plasmid in presence of BODIPY-Lys-tRNA$_{Lys}$, which enables the fluorescent labelling of translation products at lysine residue sites. Using the reported expression patterns for affinity-purified TF ensembles from pLD overexpression experiments[16], we could assign the majority of the de novo synthesised protein subunits to the to the respective TFs (Supplementary Fig. 5). To improve detection sensitivity and enable quantification of newly synthesised proteins, we also performed a mass spectrometry-based quantitative protein expression analysis using stable-isotope labelling[19]. For this purpose, we carried out PURErep in vitro experiments with each pLD plasmid with $^{15}N_2^{13}C_6$-lysine as sole source of lysine and $^{15}N_4^{13}C_6$-arginine as the sole source of arginine. Using the unlabelled PURE-supplemented TFs as internal standards to determine the heavy-to-light (H/L) ratio of

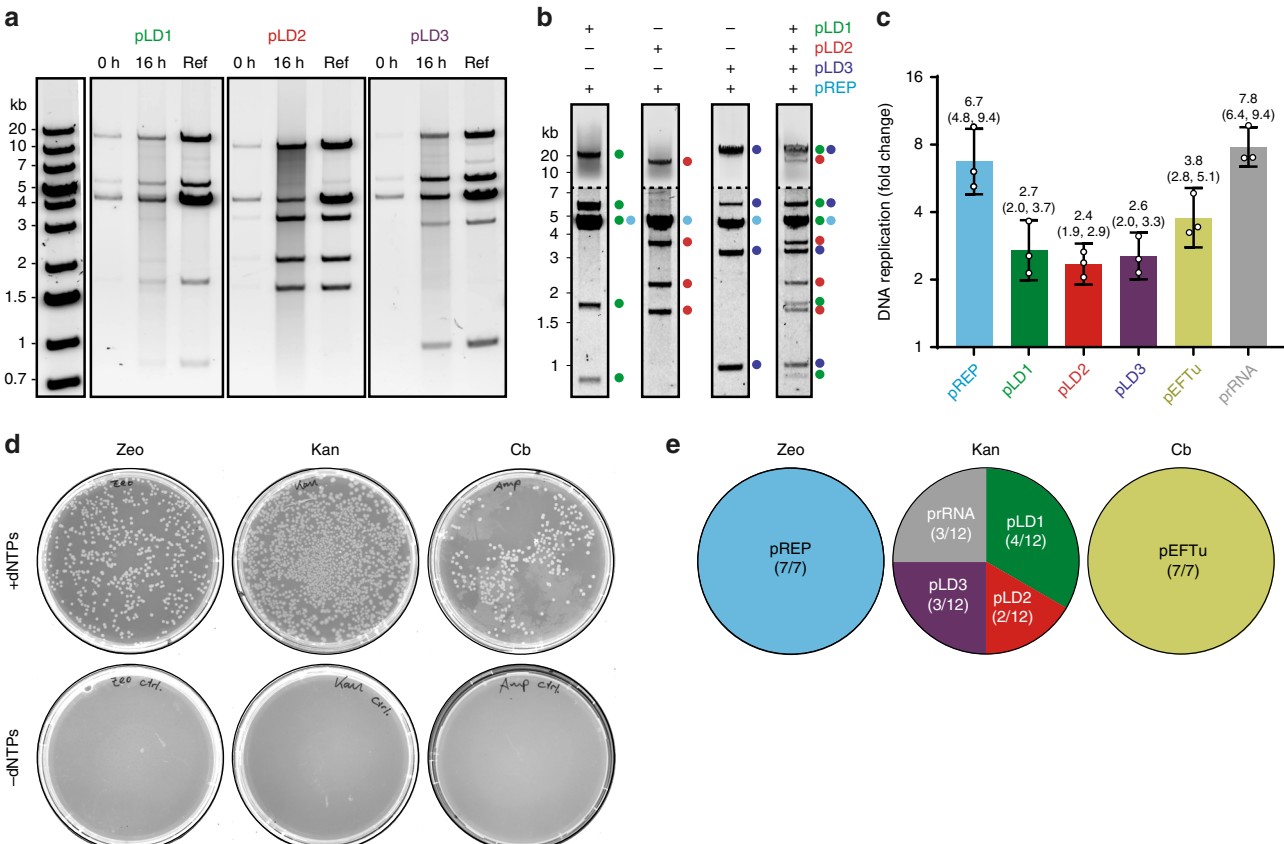

**Fig. 2 In vitro replication of large multicistronic DNA constructs. a** MluI restriction patterns of gel-purified TTcDR products from individual pLD/pREP co-replication experiments at $t = 0$ h and $t = 16$ h. Concentrations were 6 nM pREP and 0.7 nM pLD1, pLD2 or pLD3. Authentic control standards for clonal pREP/pLD mixtures are shown for each TTcDR reaction. The raw gel image is shown in Supplementary Fig. 3A. Samples were tested in three biological replicates. **b** Representative restriction digest of individual pLD co-replication experiments (lanes 1–3, 2 nM pLD plasmid, 4 nM pREP) and co-replication of all pLD plasmids (lane 4, 4 nM pREP, 2 nM pLD1-3). The specific MluI cleavage products for each plasmid are colour-labelled (pREP—cyan, pLD1—green, pLD2—red, and pLD3—purple). To improve the visibility of low-molecular-weight bands, the lower parts of the bands are presented with different contrast settings (indicated by the dotted line). The unprocessed gel images are shown in Supplementary Fig. 3B. Samples were tested in biological replicates. **c** Fold changes of the six plasmids pREP, pLD1-3, prRNA and pEFTu after an overnight TTcDR relative to their respective input concentrations determined by qPCR using plasmid-specific amplicons (shown are means with their 68% CI from biological triplicates using different PURErep batches). **d** LB plates of *E. coli* 10-beta cells transformed with overnight TTcDR reactions from **c**. TTcDR reactions were carried out in presence (+dNTPs) or, as negative control for background colonies, in absence (-dNTPs) of dNTPs. Cell-derived plasmid templates were digested with DpnI. Cells were grown under selective conditions for zeocin (Zeo, pREP), kanamycin (Kan, pLD1-3, prRNA) and carbenicillin (Cb, pEFTu). **e** Circle diagrams of the relative transformation frequencies for the six plasmid species isolated from 26 randomly picked colonies from the three +dNTP plates in d (7 for Zeo, 12 for Kan and 7 for Cb). For the restriction pattern analyses, see Supplementary Fig. 3C–E. Source data are available in the Source Data file.

isotope-labelled peptides, we found strong evidence for the de novo synthesis of all pLD-encoded TF protein subunits in overnight reactions (Fig. 4a). In particular, we obtained H/L ratios close to or larger than one for 12 of the 13 TFs encoded on pLD1 implying full regeneration of most of the encoded proteins during IVT. Partial or even full regeneration was also observed for the proteins encoded on both pLD2 and pLD3 (Fig. 4a).

Next, we probed multicistronic expression of all three pLD plasmids during parallel TTcDR induced by the addition of pREP. Despite the considerably increased synthetic burden (replication of a 78 kb genome and transcription/translation of 33 protein chains), we detected H/L ratios > 0.73 for 16 of the 32 encoded protein subunits. The H/L ratios of remaining TF subunits indicated regeneration levels between 10–70% ($N = 10$) and 4–9% ($N = 6$) (Fig. 3b). Thus, even under non-optimised batch conditions, PURErep in combination with pREP enables both the complete replication of 32 pLD-encoded TF cistrons as well as expression of about half of the encoded TF peptide chains in yields comparable or exceeding their initial PURErep input concentrations.

## Discussion

We demonstrated that under optimised TTcDR conditions, synthetic multipartite DNA genomes approaching the size of a postulated MPC genome can self-replicate and express proteins under cell-free conditions. Surprisingly, primer-free TTcDR by Phi29-DNAP alone is already sufficient to generate a significant amount of monomeric replication products from circular plasmids without any enzymatic post-processing, suggesting that partially recursive genome replication in cell-free systems can be achieved with only a single DNA polymerase. Furthermore, both the monomeric and concatemer TTcDR products can be directly transformed into *E. coli* where they propagate as authentic copies of their parental plasmid presumably after re-circularisation by intramolecular homologous recombination[20].

PURErep enables the self-encoded replication of plasmid ensembles with a total DNA length that exceeds both the size of a proposed minimal genome for a self-replicating translatome[3] and that of the smallest known bacterial genome (*Nasuia deltocephalinicola*, 112 kb)[21]. Currently, most of the space in our

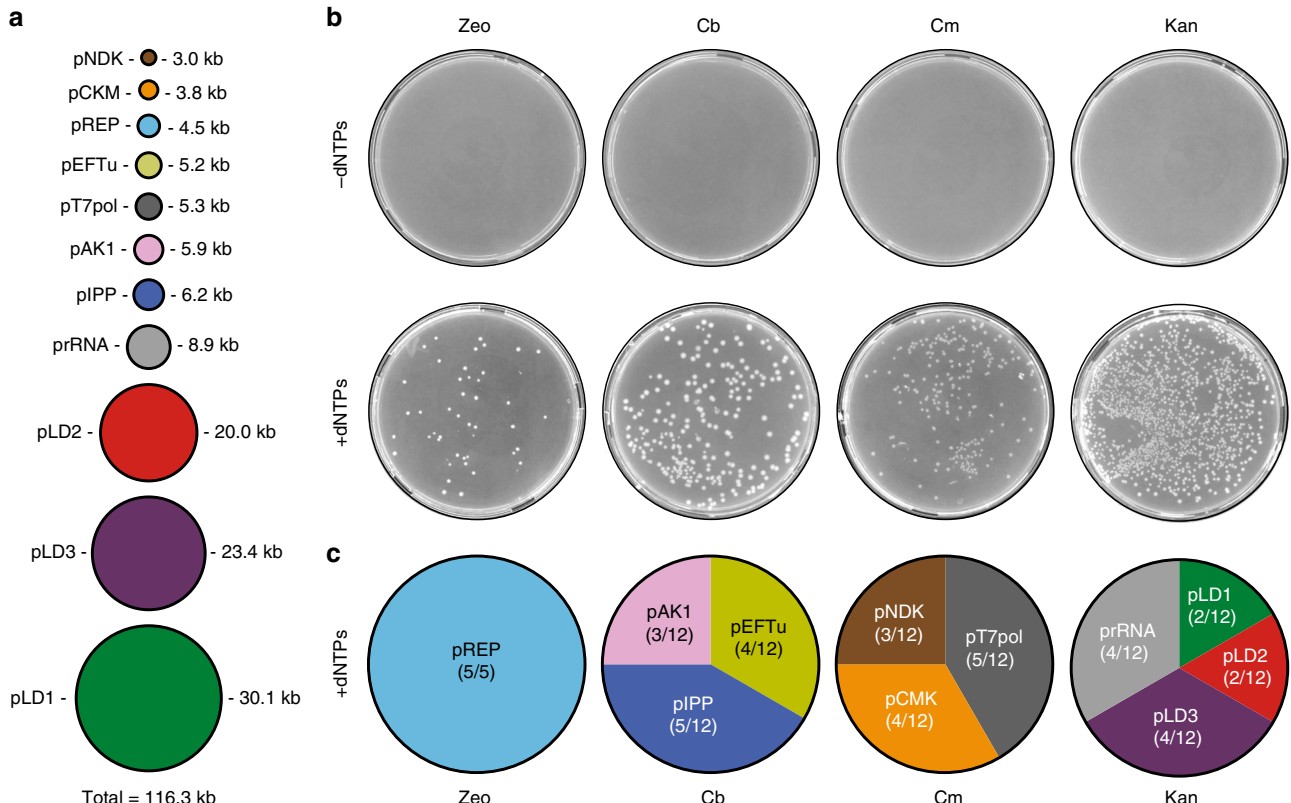

**Fig. 3 In vitro replication of a 116.3 kb multipartite DNA genome. a** The collection of 11 plasmids (combined length 116.3 kb), which were co-replicated in PURErep. In addition to Phi29-DNAP (pREP, 6 nM input), the plasmid ensemble encodes the ribosomal rRNA operon *rrnB* (prRNA, 3 nM input), all *E. coli* TFs (pLD1-3, pEFTu, 0.6 nM input each), a minimal NTP regeneration system (pNDK, pCKM, pAK1, 3 nM input each), T7-RNA polymerase (pT7pol, 3 nM input) and inorganic pyrophosphatase (pIPP, 3 nM input). **b** LB plates of *E. coli* 10-beta cells transformed with overnight TTcDR reactions containing all 11 plasmids from a. TTcDR reactions were carried out in presence (+dNTPs) or, as negative controls, in absence (−dNTPs) of dNTPs. Cell-derived plasmid templates were digested with DpnI after TTcDR. Cells were grown under selective conditions for zeocin (Zeo, pREP), carbenicillin (Cb, pAK1, pIPP, pEFTu), chloramphenicol (Cm, pNDK, pT7RNAP, pCMK) and kanamycin (Kan, pLD1-3, prRNA). **c** Circle diagrams of the relative transformation frequencies of the 11 plasmid species isolated from 41 randomly picked colonies from the four + dNTP plates in B (5 for Zeo, 12 for Cb, 12 for Cm and 12 for Kan). For the restriction pattern analyses, see Supplementary Fig. 4. Source data are available in the Source Data file.

multipartite model genome is taken up by the plasmid backbones, which maintain compatibility with in vivo propagation. In future genome designs, these sections could be replaced by the ~110 genes that are currently missing for the encoding of a complete minimal replicator dependent only on small-molecule nutrients[3].

An additional core requirement for a future minimal replicator such as a MCP is the ability to regenerate its individual protein components. In proof-of-concept batch PURErep reactions, we found that de novo synthesis of 30 of the 31 essential *E. coli* TFs can be detected after TTcDR of their encoding ~78 kb plasmid ensemble. The relative (apparent) regeneration of the TFs encoded on pLD1 was generally efficient and reached H/L ratios ≥0.8 for 10/13 TFs during co-TTcDR of all pLD plasmids and >0.9 for 12/13 TFs when only pLD1 alone was added to PURErep. In comparison, much lower regeneration levels were achieved for the TFs encoded on pLD2 and pLD3. These results correlate well with the concentrations of the individual TFs in the PURErep starting solution (Supplementary Table 2): The proteins encoded on pLD1 are the lowest concentrated TFs in PURE (approximate concentrations of 15–480 nM) and therefore readily compatible with the protein expression yields that can be achieved with current recombinant IVT systems. In contrast, the initial concentrations of pLD2-encoded TFs, which performed worst in our co-expression experiments, are much higher (approximate concentration of 0.75–3.2 μM) and therefore cannot be efficiently regenerated in the current PURErep system.

While quantification using stable-isotope labelling is considered a reliable and robust methodology to determine relative expression levels (in particular in cell-free environments)[22,23], several factors such as incomplete trypsin digestion, translational arrest, incomplete peptide labelling or low peptide counts in e.g., Arg/Lys-rich proteins may affect quantification and therefore obscure the achieved regeneration levels[22–25]. Furthermore, the current MS-based approach provides no information on the correct folding of the synthesised polypeptide chain and, thus, the actual amount of functional protein obtained during expression. Therefore, a direct functional feedback of the synthesised TFs back into IVT will be required in future experiments to assess or improve the amount of active protein that can be generated during TTcDR. Fortunately, *E. coli* tRNA synthetases, which are one of the major TF factors in IVT systems, can be very well expressed in their active soluble form in PURE[26]. Thus, it seems conceivable that the current IVT activity of PURErep is sufficient to generate systems capable of regenerating self-coded proteins of which only low concentrations are required.

In addition to addressing the functional state of the in vitro expressed proteins, self-encoded regeneration of TF factors will require further optimisation of expression stoichiometries and yields. Balanced stoichiometries could be achieved through optimised ribosome-binding sites, cistron positioning, promotor optimisations or feedback regulation[27,28]. Enhanced protein expression will most likely require continuous mode cell-free

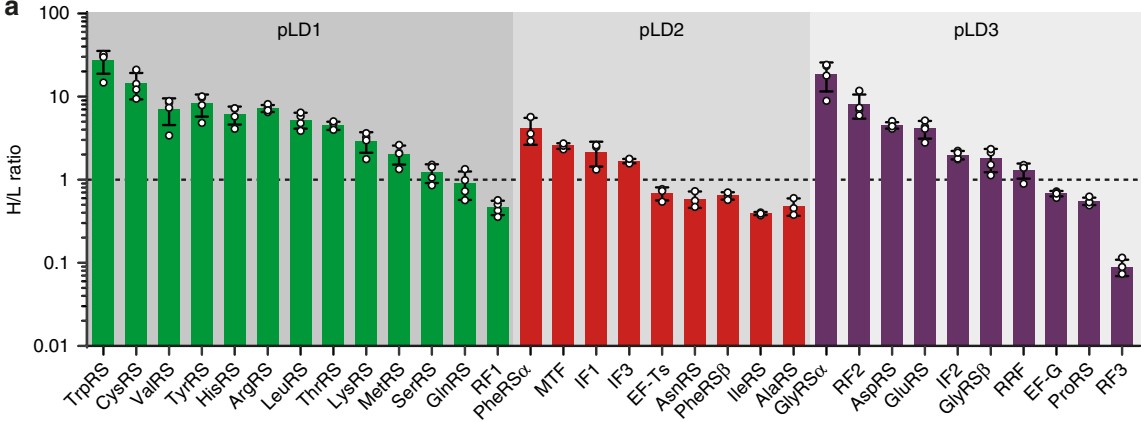

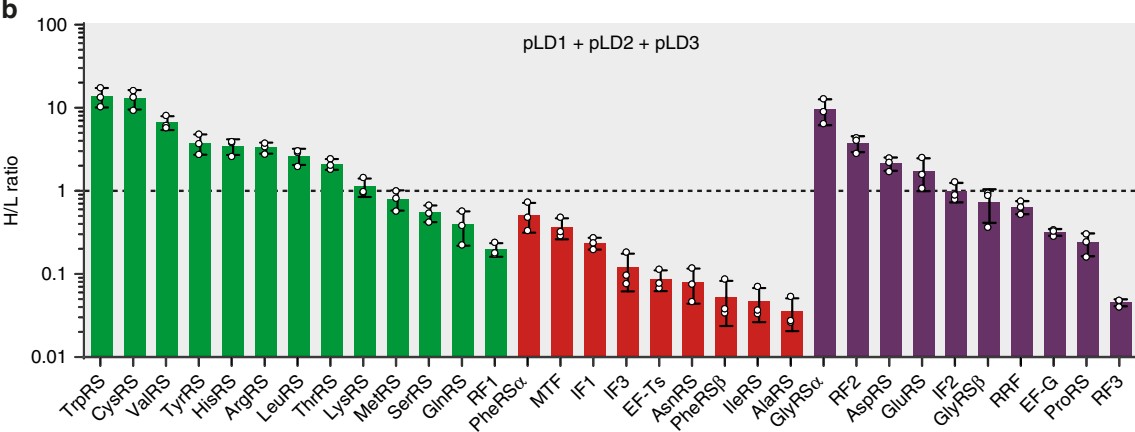

**Fig. 4 A substantial number of TFs are efficiently expressed during co-replication of pLD1, plD2, pLD3 and pREP. a** H/L ratios of the 32 TF protein subunits after $^{15}N_2^{13}C_6$-lysine and $^{15}N_4^{13}C_6$-arginine incorporation during in vitro transcription/translation of pLD1 (green, $n = 4$), pLD2 (red, $n = 3$) or pLD3 (purple, $n = 4$) in PURErep overnight reactions. The line H/L = 1 indicates full regeneration of a protein to its respective input concentration. **b** Expression levels during parallel TTcDR of pLD1 (green), pLD2 (red) and pLD3 (purple) during TTcDR induced by the addition of pREP. All H/L values are means $+/-$ s.d. ($n = 3$) from biological replicates triplicates using different PURErep batches. Source data are available in the Source Data file.

protein synthesis setups, e.g., based on miniaturised fluid array devices, which greatly increase protein yields[29]. Using this approach, the regeneration of the other pLD-TFs, EF-Tu and the other proteins of the PURE enzyme mix could be achieved. The construction a self-regenerating IVT-system that is completely independent from the external supply of external macromolecules will also require integrated ribosome synthesis, assembly and translation (iSAT)[30–32]. Recent non-commercial protocols for in-house PURE-production[16,18,33,34] provide attractive starting points for the generation of improved PURErep formulations that may be compatible with these key activities.

In its current form, PURErep can achieve modular in vitro replication of large genome-sized plasmid ensembles that retain their compatibility with bacterial in vivo propagation. This direct transferability could improve design, evolution and prototyping of MPC modules, orthogonal central dogmas[35] or synthetic gene circuits[36], which were before not amenable to TTcDR-based in vitro replication.

## Methods

**DNA constructs.** All primers used for cloning, mutagenesis and/or qPCR are listed in Supplementary Table 3, and were either ordered from IDT or Eurofins. All plasmids used in this study are listed in Supplementary Table 4. The open-reading frame of the Phi29 DNA-Polymerase (Gene ID: 6446511) was ordered as synthetic gene (gblock, IDT) and cloned into a pCR-Blunt vector using the ZeroBlunt cloning kit (Thermo Fisher) according to the manufacturer's instructions. The resulting

construct was further optimised for in vitro translation by adding a T7 promoter with T7 gene 10 translation-enhancer sequence[37] and a downstream bidirectional transcription terminator using Q5 site-directed mutagenesis (NEB) according to the supplier's instructions. The identity of the final construct pREP was verified by sequencing. All cloning procedures were performed with chemically competent *E. coli* DH5alpha. The plasmids pLD1, pLD2 and pLD3 were a generous gift from A. Forster (Uppsala University) and are described in more detail elsewhere[16]. The plasmid pEFTu, which for historic reasons also encodes a gene copy of IF-1, was constructed from respectively linearised genes and a pIVEX 2.3d backbone using the HiFi assembly kit (NEB). First, an intermediate version was assembled from linear overhang PCR products using the primer pairs 152, 152 (IF-1 fragment) and 153, 154 (pIVEX backbone). Subsequently, three linear overhang PCR products created using the primer pairs 161, 162 (EF-Tu fragment) and 163, 164 (gene spacer fragment) and 158, 159 (intermediate backbone containing IF-1) were assembled into the final construct. For the generation of prRNA, the *E. coli* ribosomal operon *rrnB* was directly amplified from Top10 *E. coli* by colony PCR using Q5-DNA polymerase with the primer pairs 85, 86, and cloned into a pCR-Blunt vector using the ZeroBlunt cloning kit (Thermo Fisher) according to the manufacturer's instructions. Plasmids encoding for nucleotide-diphosphate kinase (pNDK, ID:124136)[7], T7-RNA polymerase (pT7RNAP, ID:124138)[7], creatine kinase m-type (pCKM, ID:124134)[7], inorganic pyrophosphatase (pIPP, ID:118978)[18] and adenylate kinase 1 (pAK1, ID:118977)[18] were obtained from Addgene. Ampicillin-resistance genes were deleted by PCR in pT7RNAP, pCKM and pNDK using primers 200 and 201 (Supplementary Table 3). dsDNA concentrations were measured using a NanoDrop One-c (Thermo Scientific) following the manufacturer's instructions. All constructs were verified by sequencing (Eurofins Genomics).

**sfGFP expression.** The difference in protein synthesis yields between PURExpress and PURErep was estimated using fluorescence of de novo synthesised sfGFP. To this end, 25 μL PURExpress reactions were set up according to the manufacturer's

instructions using 150 ng of pIVEX-sfGFP plasmid. In total, 25 μL PURErep reactions consisted of 2.5 μL 10× energy mix (EM, Supplementary Table 1), 1 μL solution A (PURExpress, NEB), 15 μL solution B (PURExpress, NEB), 0.6 μL 25 mM dNTPs (equimolar), 0.5 μL rNTP mix (18.75 mM ATP, 12.5 mM GTP, 6.25 mM UTP/CTP) and 150 ng pIVEX-sfGFP plasmid DNA. After 2 h of incubation at 37 °C, 2× SDS loading buffer was added to the respective mixtures, and they were incubated for 5 min at 55 °C to preserve sfGFP fluorescence. In all, 10 μL of each sample was subsequently loaded on a 12% polyacrylamide SDS-Gel. Fluorescent bands were directly visualised using a Typhoon FLA 7000, and analysed via ImageQuant, GE Healthcare Life Sciences. To assess pLD-plasmid encoded gene expression, pLD plasmids (final concentration 4 nM) were added to PURErep reactions containing 1 μL of FluoroTect GreenLys (Promega). Prior to the addition of 2× SDS loading buffer, the samples were incubated for 30 min at 37 °C with 1 μL of RNase Cocktail (Thermo Fisher Scientific). After denaturing PAGE, de novo synthesised proteins were visualised using a Typhoon FLA 7000 scanner.

**Transcription–translation-coupled DNA replication**. The reaction composition for a typical 25 μL TTcDR reaction was as follows: 2.5 μL 10× EM, 1 μL solution A (PURExpress, NEB), 15 μL solution B (PURExpress, NEB), 0.6 μL 25 mM dNTPs (equimolar), 0.5 μL rNTP mix (18.75 mM ATP, 12.5 mM GTP, 6.25 mM UTP/CTP) and plasmid DNA (as specified in the main text). If necessary, the reaction volumes were adjusted to 25 μL with ddH₂O. TTcDR reactions in the conventional PURExpress system were assembled according to the standard protocol for a 25 μL reaction: 10 μL solution A, 7.5 μL solution B, 0.6 μL 25 mM dNTPs, plasmid DNA as specified in the main text. The final reaction volume was adjusted to 25 μL with ddH₂O. PURExpress and PURErep samples were incubated at 30 °C for up to 16 h in a ProFlex thermocycler (Applied Biosystems) for the time indicated. Time point zero samples were aliquoted directly after mixing, flash-frozen in liquid nitrogen and stored at −80 °C until further use.

**Gel analysis of TTcDR products**. Untreated TTcDR samples were directly analysed by neutral agarose gel electrophoresis in 1× TAE (Tris-Acetate-EDTA, gels pre-stained with SYBR-safe). Due to the size of some rolling-circle concatemers and/or due to the possible formation of MgPPi-DNA nanoparticles[15], a fraction of the total product remained in the gel pockets. When defined product bands were desired, such as in Figs. 1c and 2a, samples were treated with FastDigest MluI (Thermo Fisher Scientific, simply referred to as MluI throughout the rest of the paper) in 1 × FastDigest buffer according to the manufacturer's instructions. Gels were imaged using a Typhoon FLA 7000, GE Healthcare Life Sciences using the Typhoon Scanner Control 5.0 software package and analysed using ImageQuant TL 8.1 and/or ImageJ 1.51i. To confirm the identity of the replication products pLD1-3 by restriction pattern analysis, TTcDR samples were processed by adding 1 μL RNAse Cocktail (Thermo Fisher) and 1 mg/ml Proteinase K. After 16 h of incubation at 37 °C, the samples were loaded on a neutral 0.8% agarose gel. DNA products migrating at a size of ~20–30 kb were gel-extracted and purified using the Zymoclean Large DNA Fragment Extraction Kit (Zymo Research). Purified DNA was cut with MluI and restriction patterns visualised by neutral gel electrophoresis as described above. The reference lanes consisted of purified plasmids (or mixtures thereof), digested with MluI.

**In vivo propagation of TTcDR products**. For transformation experiments, TTcDR samples were digested with Dpn1 (NEB) (1.5 h at 37 °C) to remove parental plasmid DNA before transforming 2 μL of the mixture into 50 μL electro-competent 10-beta E. coli cells (NEB). Transformants were selected on LB-agar plates supplemented with either zeocin (pREP), carbenicillin (pEFTu, pIPP, pAK1), chloramphenicol (pNDK, pT7RNAP, pCKM) or kanamycin (pLD plasmids, prRNA). Fingerprints restriction digests of plasmids isolated from cells grown in presence of zeocin and kanamycin were performed with MluI as described in the previous section. Plasmids isolated from chloramphenicol plates were digested with XbaI (NEB). Plasmids isolated from carbenicillin plates were either digested with EcoRV (NEB) or XbaI as indicated in the respective figure legends. In vivo propagated TTcDR products and plasmids were prepared from overnight E. coli cultures using the NucleoSpin Plasmid kit (Macherey Nagel) following the manufacturer's instructions.

**Relative DNA quantification by qPCR**. Fold changes of DNA copy-number relative to input levels ($t = 0$) were measured by qPCR (Luna Universal Mix, NEB) in a StepOne Real-Time PCR System (Thermo Fisher Scientific, StepOne/StepOnePlus Software v2.3). For each time point, three individual samples were taken and diluted 4000-fold in ddH₂O, which were further diluted 1:20 in the final qPCR reaction (final dilution 1:80,000). The specific primers for each target amplicon are listed in Supplementary Table 3. The fold change $f$ at time point $t$ was calculated using the equation:

$$f(t) = E^{\Delta C_q(t)}$$

where $f(t)$ is the fold change of the sample at time point $t$, $E$ the PCR efficiency and $\Delta C_q(t)$ the average difference between the qPCR cycle thresholds $\Delta C_q$ at time zero

and time $t$. $E$, $C_q(0)$ and $C_q(t)$ were determined using LinRegPCR[38] (version 2018.0). Different TTcDR time points of the same experiment were quantified in the same qPCR experiment using a common primer/enzyme mastermix for each target plasmid. Asymmetric upper and lower confidence limits (68%) for $f(t)$ were approximated by calculating $f(t)$ for $\Delta C_q(t)$ + s.d. and $\Delta C_q(t)$ − s.d., respectively, where s.d. is the standard deviation for $\Delta C_q(t)$ values from replicates as stated in the respective figure legends (typically $n = 3$). All data sets were visualised using Graphpad Prism 7.05.

**Stable-isotope labelling of co-expression products**. For stable-isotope labelling of de novo synthesised protein, TTcDR samples were mixed with an energy mix containing ¹⁵N₂¹³C₆-lysine and ¹⁵N₄¹³C₆-arginine, instead of the corresponding unlabelled amino acids. After incubating for 2 h at 37 °C, the samples were analysed via mass spectrometry. First, the reaction mixture was diluted with equal volumes of buffer containing 1% sodium deoxycholate, 10 mM TCEP and 40 mM chloroacetamide in 25 mM Tris•HCl at pH 8.5 to be incubated at 37 °C for 20 min. The reaction mixture was further diluted and incubated overnight with roughly 1 μg of trypsin. Digested peptides were acidified and purified through SCX (strong cation exchange) StageTips (Thermo Scientific). Liquid chromatography–mass spectrometry (LC-MS) analysis was performed on a Q-Exactive-HF mass spectrometer (Thermo Scientific) operated in a data-dependent fashion. The raw data were processed using the MaxQuant[39] computational platform (version 1.6.6.0), and all peptide and protein identifications were filtered at 1% false discovery rate. The derived peak list was searched using Andromeda search engine integrated in MaxQuant against the E. coli K12 proteome (Proteome ID: UP000000625/Genome accession: U00096) obtained from UniProt (4391 protein entries; last modified May 14 2019). The obtained H/L values for each pLD-encoded protein were analysed and plotted using Graphpad Prism 7.05.

**Reporting summary**. Further information on research design is available in the Nature Research Reporting Summary linked to this article.

## Data availability

The source data underlying Figs. 1b, 1c, 1e, 1f, 2a, 2b, 2c, 2e, 3b, 4a, 4b, Supplementary Figs. S1B, S1C and 5 are provided in a Source Data file. MaxQuant outputs are provided in two separate source files. Plasmids encoding for nucleotide-diphosphate kinase (ID:124136), T7-RNA polymerase (ID:124138), creatine kinase m-type (ID:124134), inorganic pyrophosphatase (ID:118978) and adenylate kinase 1 (ID:118977) were obtained from Addgene. Further data supporting the findings of this paper are available from the corresponding author upon reasonable request.

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

## Acknowledgements

The authors especially thank A. Forster (Uppsala University) for providing the plasmids pLD1, pLD2 and pLD3 to the community. We thank N. Nagaraj and the MPIB mass spectrometry core facility for the help with the mass spectrometry data acquisition and analysis. We thank P. Schwille and L. Kei for constructive discussions and comments. Special thanks to K. Le Vay for providing critical comments and reviewing of the paper. Funding was provided by the MaxSynBio consortium, which is jointly funded by the Federal Ministry of Education and Research of Germany and the Max Planck Society. M. Heymann gratefully acknowledges support from the Joachim Herz Foundation.

## Author contributions

H.M. conceived the project. K.L. and H.M. designed the experiments. M.H. contributed to plasmid sample preparation, discussed the results and commented on the paper. K.L. carried out all experiments. R.H. contributed to qPCR experiments. K.L. and H. M. wrote the paper.

## Competing interests

The authors declare no competing interests.
