## [Peer Review File · Nature Communications]

Reviewers' Comments:

Reviewer #1:

Remarks to the Author:

This manuscript by Libicher et al represents an advance by describing the improvement of the established PURE system to support in vitro replication of several plasmids with a combined size amounting to about 90 kb. Moreover, in vitro transcription and translation is shown from this replicated plasmid ensemble.

To achieve these results, the authors started from the PUREexpress system, and altered its composition with the objective to optimize replication of the plasmids they supplemented. This was inefficient using previously available protocols and is now feasible with their here described "PURErep" protocol.

More precisely, the authors altered the composition of the reaction mixture by supplying translation factors, ribosomes and reducing agents, while concomitantly decreasing tRNA and rNTP levels which had earlier been found to be detrimental to replication.

The manuscript is well written, properly illustrated and adequately referenced. The results presented are sound and stand on their own. The data supports the conclusions drawn and evidences that the PURErep system is efficient in transcription-translation coupled DNA replication. The authors show expression of about half of the E.coli transcription factors as an example for the efficacy of their system.

The study presented demonstrates that efficient in vitro replication of several plasmids can be achieved by starting from an established in vitro transcription/translation system, carefully supplementing and gauging components of the system and optimizing the protocols used. The resulting protocol the authors present will be useful for a range of studies investigating replication, transcription and translation in vitro, notably in minimal synthetic cell systems - an addition to the tool-box that will be appreciated within the community. Thus, the work presented here has clear merit. The authors point out in their Discussion that the system and protocols they developed will enable a range of such applications. It would add to the manuscript if the authors presented a meaningful application of the kind they allude to, beyond expressing E.coli transcription factor proteins from plasmids.

Reviewer #2:

Remarks to the Author:

The manuscript is focused on developing an in vitro translation system based on the well-known PURE system to have self-encoded replication. The motivation is founded on the interest of cell-free reconstitution of the central dogma to include DNA replication that is of high interest. The approach for transcription and translation coupled DNA replication (TTcDR) is based on circular DNA encoding phi29 DNA polymerase, which is the same/similar approach published by Yomo and coworkers (Sakatani et al.). The authors reported optimization of the PURE system to improve DNA polymerase activity by lowering tRNA and rNTP concentrations and showed that it can complete replication of several plasmids in a one-pot reaction. Using this system, the authors demonstrated co-replication of 6 plasmids (totaling 93 kb), that include 82% of the predicted genome length to encode all the essential genes to realize the central dogma in vitro. The authors used mass spectrometry to analyze protein expression and claimed that the TTcDR synthesized 30 TFs and with 15 in amounts that are comparable to other published PURE systems. Generally, the manuscript is clearly written, but the advance presented this work can be a bit subjective. This does move the field towards semi-autonomous reproduction – at least with an in vitro system that has DNA replication, transcription, and translation – but I feel that the work fell short in terms of the advance presented that would move the dial. I also have some concerns regarding the experimental design and conclusions.

Major comments:

- In many of the presented data, the authors presented technical replicates (e.g. Fig 1B, 1E,

1F...etc). It is well known that that PURE systems can vary from batch to batch and it is not clear how robust are the data reported here from biological replicate as the authors make no mention of the measure they have taken to ensure reproducibility. Many in the field remark on this and this has been reported by Danelon and coworkers. In general, biological replicates are necessary and should be presented as a true measure of variability. When technical replicates are presented, one cannot really draw meaningful conclusions. I am a bit surprised if only technical triplicates were performed for these experiments. It would be necessary to capture the variability in a more rigorous manner, in my opinion. I am also wary of the error analysis/presentation itself. Many of the data are presented as fold-change in a log scale. For instead in 1E, the error bars presented for 6 hour could range from 30 to 100 fold. Is that really what was measured? In some cases, authors may wish to present the error in actual numbers on top of the bar graph (for instance for 1B). The statement that PURErep yielded 24% lower GFP expression compared to standard PUREexpress is also weakly supported by the data presented in the supplemental data (not a very quantitative comparison). The conclusion on a modest reduction on protein expression strength should be better supported.

- The fold-change data presented in Figure 2C does not coincide with the text stating that a fold change of 2-8 fold was observed (pREP was close to 30 and pLD3 is probably 10). As mentioned in the previous point, the error representation is also questionable for all the conditions. The authors stated qPCR experiments 'implied' that all six plasmids were, what implied? I also find the description of this experiment to be quite vague, even after looking at the methods section. Are there any internal controls or normalization run for this? What is the fold change compared with (i.e. what is the 1 fold representing here)?

- Mass-spec experiment on TF synthesis. Are there any reasons why 2 of the TFs were not detected at all? I don't see any mention of this and should be discussed. One of the TFs seems to be from the construct that is generated from this work. Is there any validation that this TF actual can be expressed or is this an instrumentation/detection issue? My main concern with the data presented is the interpretation. The authors used the term full regeneration for H/L over 1 (only 7 out of 30 TFs). To me, this data doesn't seem so impressive. Ratios lower than 1 (between 0.1-0.9) were claimed to be partially regenerated (another 15). I think more discussion would be needed that center around this data. Would the goal to be regenerate TFs in the same amounts as PURE? If so, we are probably more than a few steps away from this. I presume TF that is initially present in PURErep is not degraded, is that correct? Figure 3B basically reports how the PURErep system compares to others (basically presenting the data from A differently). What I had troubles with is to draw the conclusion that 15 TFs are made at amounts comparable to other systems, but not pointing out about others. It was clear from the data that many of the TFs produced in PURErep syn fell far short of that of PURErep, making this comparison a bit odd in my opinion. PURE v1, 2, 7 from Kazuta et al. reported significant variations in protein synthesis yield and other differences, with v1 and v2 far worse than v7. This is like comparing apples and oranges and I have a hard time with the low amounts of produced TFs and stating this is comparable to other systems when other systems have different performance metrics. The estimate of the concentration comes from the mass spec data and the estimate of what the PURErep has. It would be prudent for the authors to describe potential sources of errors in this estimate. To me, it would be great if the authors have some additional independent data on measuring the concentrations of some of these produced TFs via other approaches to validate/corroborate the mass spec results.

Other comments:

- Abstract: The last sentence in the abstract is not well supported. I guess it can be on how reads 'conclude' in this context. I did not see any experimental evidence or in-depth discussion on what supports that only a few further steps are required to generate semi-autonomous reproduction.
- What is the dotted line across the blot in Figure 2B?

Minor comments:

- Line 27: minimal protein-based cells

- Line 82: why linear?
- Line 88: italicize *E. coli*.
- Line 118: mass spectrometry-based
- Line 131: reference needed
- Line 146: self-regeneration of significant fraction of TF proteins. This statement can be more precise as this is from the result of the work.
- Line 155-156: this is an awkward sentence and require revision.
- Line 197-198: this sentence doesn't make sense to me, and what is formyl-donor 10-formyl-5,6,7,8-THF?
- Line 201: sentence is incomplete
- Line 233: Mlu1 was mentioned earlier and while knowing it is a restriction enzyme, it may not be clear to other the samples are treated by FD-Mlu1 in this sentence. This is minor but may be helpful to provide rationale for reader.
- Line 244: remove 'with' after were

Reviewer #1 (Remarks to the Author):

1.1. (...) Thus, the work presented here has clear merit. The authors point out in their Discussion that the system and protocols they developed will enable a range of such applications. It would add to the manuscript if the authors presented a meaningful application of the kind they allude to, beyond expressing E.coli transcription factor proteins from plasmids.

We thank the reviewer for this positive comment. We have now expanded coding potential of our system and replicated genomes that also include modules of a minimal transcription machinery (T7 polymerase) as well as a minimal energy regeneration system based on creatine kinase, adenylate kinase, nucleoside disphosphate kinase and pyrophosphatase. Together with the other co-replicated genes, we are now able to self-replicate a genome that contains all but one cistrons of the PURE enzyme mix together with all three ribosomal rRNAs and the Phi29 DNA polymerase. With respect to the anticipated applications, which we briefly discuss at the end of the manuscript (e.g. evolution of (new) MPC modules, orthogonal central dogmas or synthetic gene circuits): We believe that establishing / carrying out these experiments in a meaningful manner will require a considerable amount of time and therefore go beyond the scope of this manuscript.

Reviewer #2 (Remarks to the Author):

Major comments:

2.1. *In many of the presented data, the authors presented technical replicates (e.g. Fig 1B, 1E, 1F....etc). It is well known that that PURE systems can vary from batch to batch and it is not clear how robust are the data reported here from biological replicate as the authors make no mention of the measure they have taken to ensure reproducibility. Many in the field remark on this and this has been reported by Danelon and coworkers. In general, biological replicates are necessary and should be presented as a true measure of variability. When technical replicates are presented, one cannot really draw meaningful conclusions. I am a bit surprised if only technical triplicates were performed for these experiments. It would be necessary to capture the variability in a more rigorous manner, in my opinion.*

We thank the referee for this helpful suggestion. We have now repeated all TTcDR experiments as “biological” replicates using different PURE batches. During these experiments, we have also further optimized our qPCR protocol to improve reproducibility (see next comment). Biological triplicates are now available for Fig. 1B (DNA replication in PURExpress vs PURErep), Fig. 1E (replication kinetics with different pREP input concentrations), Fig. 2C (co-replication of six plasmids), Fig. 4A (expression experiments of the individual pLD-plasmids in PURErep) and Fig. 4B (TF expression from all pLD-plasmids during TTcDR). The only exception is the proof-of-concept serial transfer experiment Fig. 1F, which was carried out in technical replicates using the same PURE-batch due to the high expenses associated with the PURExpress system, which forms the basis of our current version of PURErep.

2.2. *I am also wary of the error analysis/presentation itself. Many of the data are presented as fold-change in a log scale. For instead in 1E, the error bars presented for 6 hour could range from 30 to 100 fold. Is that really what was measured?*

We thank the referee for this comment. The fold change (fc) is an estimate derived from the C_q-values and calculated by $fc = E^{\Delta C_q}$, where ΔC_q is the difference between the C_q-value of the qPCR traces for the same amplicon at t(0) and e.g. t(6 h) and E the qPCR efficiency (ideally ~2). From this equation, it is obvious that small changes in C_q can lead to drastic changes in fc. Similar error-ranges were also reported for other papers in the field, where fold change of DNA was measured during TTcDR using in qPCR (e.g. compare Figure 3 in <https://www.nature.com/articles/s41598-018-31585-1>). We are aware of the fact that having an internal reference (and then calculating $\Delta\Delta C_q$ -values instead of ΔC_q) might have further improved the robustness of our measurements. However, given the fact that the Phi29 DNA polymerase replicates all added DNA during TTcDR, the use of an internal standard was not possible.

Given that the estimated fold change does not follow a normal distribution due to the non-linear dependency on ΔC_q , it is not meaningful to calculate an apparent standard deviation for fc using the standard deviations determined from the Ct-values. For this reason, we estimated confidence intervals (68%) from the standard deviations of the ΔC_t values determined by error propagation: $[E^{\Delta C_q + \text{stdev}}, E^{\Delta C_q - \text{stdev}}]$. This approach leads to fairly large “error bars”. For example, assuming an ideal E = 2 and a $\Delta C_t = 6 \pm 1$ we get a fc of 64 with a confidence interval of (32, 128). We realised that we

mislabelled the figure legends by writing “(mean ± s.d.)” where in fact we did not show the standard deviations but rather the confidence intervals for the estimated fc. We apologise for this mistake.

Nevertheless, we agree with the referee that the large confidence intervals were a bit worrying. Therefore, we further optimized our qPCR assay to improve repeatability and obtain lower variances. Using a higher dilution of the TTcDR samples in the qPCR reactions, we were able to considerably improve reproducibility of the experiments while also keeping the confidence interval for biological triplicates at an acceptable level (e.g. Fig. 1B or 2C). We now get very reproducible fold-changes for pREP throughout the manuscript in the different biologically independent replicates (e.g. Figure 1B vs 1E). Using the new experimental protocol, we could also drastically decrease the error bars for the co-replication experiment in Fig. 2C.

We hope that the referee appreciates our efforts in improving the statistics and reproducibility of our experiments.

2.3. In some cases, authors may wish to present the error in actual numbers on top of the bar graph (for instance for 1B).

We thank the referee for this helpful suggestion. We have now added the numbers on top of the bar graphs shown in Fig. 1B and Fig. 2C.

2.4. The statement that PURErep yielded 24% lower GFP expression compared to standard PUREexpress is also weakly supported by the data presented in the supplemental data (not a very quantitative comparison). The conclusion on a modest reduction on protein expression strength should be better supported.

We thank the referee for this suggestion. We have now performed additional in-gel quantifications of sfGFP expression in biological triplicates. The new data is now presented in Supplementary Fig. 1 and supports our previous finding that protein expression is only modestly reduced in PURErep.

2.5. The fold-change data presented in Figure 2C does not coincide with the text stating that a fold change of 2-8 fold was observed (pREP was close to 30 and pLD3 is probably 10). As mentioned in the previous point, the error representation is also questionable for all the conditions.

We apologise for this mistake. In the new version of the manuscript, we have now updated the figures with the new qPCR data and reference the presented values appropriately in the main text.

2.6. The authors stated qPCR experiments ‘implied’ that all six plasmids were, what implied? I also find the description of this experiment to be quite vague, even after looking at the methods section. Are there any internal controls or normalization run for this? What is the fold change compared with (i.e. what is the 1 fold representing here)?

The sentence now reads: ***“qPCR experiments targeting plasmid-specific amplicons confirmed that monomer units of all six plasmids (total DNA length 93 kb) were co-replicated about 2-8 fold relative to their respective input levels in the presence of pREP and dNTPs after overnight incubation (Fig. 2C).”*** For all qPCR experiments, 1-fold represents the input levels of the DNA at the beginning of the reaction. We have also updated the methods section and all figure legends to make this more obvious.

2.7. Mass-spec experiment on TF synthesis. Are there any reasons why 2 of the TFs were not detected at all? I don't see any mention of this and should be discussed.

We realised that our initial MS-approach in the previous submission suffered from some sensitivity issues. This resulted mainly from the fact that we only used a single stable isotope labelled amino acid (Lys(8)) and the endoproteinase Lys-C for protein cleavage. Lys-C cleaves proteins on the C-terminal side of lysine residues, which for lysine poor proteins can result in very few large peptides with a very poor intrinsic peptide detectability (“flyability”), typically because their ionization efficiency is very low. We have now redone all experiments (also in biological replicates) using an improved labelling approach based on the standard SILAC protocol, which uses heavy lysine and arginine labelling in combination with trypsin digestion. Since trypsin predominantly cleaves proteins at the C-terminal side of lysine and arginine, we now have much better fingerprint statistics for each protein. Using this approach, we can now reproducibly detect and quantify all of the encoded TFs. The new data is now presented in Figure 4.

2.8. One of the TFs seems to be from the construct that is generated from this work. Is there any validation that this TF actual can be expressed or is this an instrumentation/detection issue?

We apologise if our manuscript was not clear enough in this part of the results section, but there seems to be a misunderstanding: The presented co-expression / co-replication experiments for the MS-detection contain the three large plasmids pLD1, pLD2, pLD3 and pREP but not pEFTu. The main reason why plasmid pEFTu was never present in these experiments is that EF-Tu is generally required in very high amounts in PURE systems, which can currently not be synthesised in standard batch setups. For this reasons, we mainly focussed on the expression of the pLD-encoded TFs as most of them are required at more “feasible” low μM or nM concentrations. We have now made this clearer in the respective results section by pointing out that pEFTu was omitted from these experiments. However, regardless of this, we know that EF-Tu can be well expressed from this construct, as we regularly use it for recombinant expression of this protein.

2.9. *My main concern with the data presented is the interpretation. The authors used the term full regeneration for H/L over 1 (only 7 out of 30 TFs). To me, this data doesn't seem so impressive. Ratios lower than 1 (between 0.1-0.9) were claimed to be partially regenerated (another 15). I think more discussion would be needed that center around this data. Would the goal to be regenerate TFs in the same amounts as PURE? If so, we are probably more than a few steps away from this.*

Using the new SILAC-derived MS approach, we can now reliably confirm that ~15 of the TFs can be regenerated, not just seven. Moreover, if we look at the expression from the individual pLD-plasmids alone, we found that e.g. for pLD1 11 of the 12 encoded TF protein subunits can be regenerated (of course it remains to be determined if these *in situ* produced proteins are active or not). However, we agree that we should have discussed these results in more detail. Therefore, we have added a new section in the discussion part (page 7, last paragraph) where we put the MS results in the context of a potential future self-regenerating PURE system and also discuss the additional steps that are required to achieve this (i.e. proper TF-regeneration).

2.10. *I presume TF that is initially present in PURErep is not degraded, is that correct?*

Yes, that is correct. We have no indication that there is significant degradation of the TFs in PURErep or any other PURE system (e.g. from different SDS-gel analyses of long-term PURE reactions).

2.11. *Figure 3B basically reports how the PURErep system compares to others (basically presenting the data from A differently). What I had troubles with is to draw the conclusion that 15 TFs are made at amounts comparable to other systems, but not pointing out about others. It was clear from the data that many of the TFs produced in PURErep syn fell far short of that of PURErep, making this comparison a bit odd in my opinion.*

After some thought, we have come to the conclusion that the reviewer is right and that the figure is rather confusing and does not really add value to the manuscript. We have therefore removed it.

2.12. *PURE v1, 2, 7 from Kazuta et al. reported significant variations in protein synthesis yield and other differences, with v1 and v2 far worse than v7. This is like comparing apples and oranges and I have a hard time with the low amounts of produced TFs and stating this is comparable to other systems when other systems have different performance metrics.*

The main point of this figure was to show that the TF levels generated in PURErep are probably sufficient for some *in vitro* translational activity (at least based on the concentrations of the other active PURE systems), although most of the TF-levels generated in PURErep syn remain below the necessary regeneration levels required to achieve the PURErep TF input concentrations. As stated in response to comment 2.11, we have now removed this figure from the manuscript.

2.13. *The estimate of the concentration comes from the mass spec data and the estimate of what the PURErep has. It would be prudent for the authors to describe potential sources of errors in this estimate. To me, it would be great if the authors have some additional independent data on measuring the concentrations of some of these produced TFs via other approaches to validate/corroborate the mass spec results.*

We thank the referee for this helpful comment. We have now added a new section in the discussion, where we describe potential sources of error for the reported H/L values (p. 8, second paragraph). Furthermore, we have now added new semi-quantitative data on the *in vitro* expression of the TFs from the individual pLD plasmids in PURErep using the GreenLys labelling methodology (shown in the new Supplementary Fig. 5). Using the reported TF sizes and the published band patterns reported by Sheperd et al. (www.ncbi.nlm.nih.gov/pmc/articles/PMC5737471), we were able assign almost all of the bands to their respective TFs. Using this method it is unfortunately not possible to determine relative expression levels since the input TFs remain invisible. However, we believe that this additional data support the results obtained by MS (i.e. the proteins are produced and are mostly full-length). Furthermore, we are confident that the improved double labelling strategy combined with the improved trypsin digest protocol has greatly improved the MS quantification of the individual TFs.

Other comments:

- Abstract: The last sentence in the abstract is not well supported. I guess it can be on how reads 'conclude' in this context. I did not see any experimental evidence or in-depth discussion on what supports that only a few further steps are required to generate semi-autonomous reproduction.

We have now rewritten the second part of the abstract and weakened the statement made in the last sentence.

- What is the dotted line across the blot in Figure 2B?

The bands in Fi. 2B are composed of two different contrast settings of the same gel to improve the visibility of the low-molecular bands of the restriction digests. The border between the two contrast areas was made visible with the dotted lines. We have now added this information to the legend.

Minor comments:

- Line 27: minimal protein-based cells

Fixed.

- Line 82: why linear?

Fixed. The sentence now reads: "Unexpectedly, we also observed formation of ~5 kb products in unprocessed samples suggesting that TTcDR reactions may produce considerable amounts of monomeric pREP copies (Supplementary Fig. 1C)."

- Line 88: italicize *E. coli*.

Fixed.

- Line 118: mass spectrometry-based

Fixed.

- Line 131: reference needed

Fixed.

- Line 146: self-regeneration of significant fraction of TF proteins. This statement can be more precise as this is from the result of the work.

We have now rewritten this part of the manuscript and discuss now in more detail the regeneration results (in particular in the context of the different PURE TF concentrations. We have also amended results section on page 6/7 to give a better description of the data.

- Line 155-156: this is an awkward sentence and require revision.

The entire section has been revised.

- Line 197-198: this sentence doesn't make sense to me, and what is formyl-donor 10-formyl-5,6,7,8-THF?

The sentence was corrected.

- Line 201: sentence is incomplete

Fixed.

- Line 233: Mlu1 was mentioned earlier and while knowing it is a restriction enzyme, it may not be clear to other the samples are treated by FD-Mlu1 in this sentence. This is minor but may be helpful to provide rationale for reader.

Thank you for spotting this. Now fixed.

- Line 244: remove 'with' after were

Fixed.

Reviewers' Comments:

Reviewer #2:

Remarks to the Author:

The revision has greatly improved and the authors have adequately addressed all my comments. I especially appreciated the authors' effort to repeat many of the experiments to show biological replicates and the optimization of qPCR assay to improve reproducibility. The explanation of the statistics is complete – my apology for missing the confidence interval description in the methods, but I was in part confused by the figure caption in the initial submission. Repeating MS experiments using two labeled amino acids instead of one also improves the data. It's helpful to show GreenLys labeling experiment to confirm the expression of TFs. Overall, this was an excellent revision. The findings will be of interest to the field and represent an advance. I support publication.